# Machine Learning Models Predicting Cardiovascular and Renal Outcomes and Mortality in Patients with Hyperkalemia

**DOI:** 10.3390/nu14214614

**Published:** 2022-11-03

**Authors:** Eiichiro Kanda, Suguru Okami, Shun Kohsaka, Masafumi Okada, Xiaojun Ma, Takeshi Kimura, Koichi Shirakawa, Toshitaka Yajima

**Affiliations:** 1Medical Science, Kawasaki Medical School, 577 Matsushima, Kurashiki, Okayama 701-0192, Japan; 2Cardiovascular, Renal, and Metabolism, Medical Affairs, AstraZeneca K.K., Tower B Grand Front Osaka, 3-1 Ofukacho, Kita-ku, Osaka 530-0011, Japan; 3Department of Cardiology, Keio University School of Medicine, 35 Shinanomachi, Shinjyuku-ku, Tokyo 160-8582, Japan; 4IQVIA Solution Japan K.K., Keikyu dai-ichi Building, 4-10-18 Takanawa, Minato-ku, Tokyo 108-0074, Japan; 5Real World Data Co., Ltd., 76 Nakanocho, Nakagyo-ku, Kyoto 604-0086, Japan

**Keywords:** artificial intelligence, chronic kidney disease, congestive heart failure, hyperkalemia

## Abstract

Hyperkalemia is associated with increased risks of mortality and adverse clinical outcomes. The treatment of hyperkalemia often leads to the discontinuation or restriction of beneficial but potassium-increasing therapy such as renin-angiotensin-aldosterone inhibitors (RAASi) and high-potassium diet including fruits and vegetables. To date, limited evidence is available for personalized risk evaluation in this heterogeneous and multifactorial pathophysiological condition. We developed risk prediction models using extreme gradient boosting (XGB), multiple logistic regression (LR), and deep neural network. Models were derived from a retrospective cohort of hyperkalemic patients with either heart failure or chronic kidney disease stage ≥3a from a Japanese nationwide database (1 April 2008–30 September 2018). Studied outcomes included all-cause death, renal replacement therapy introduction (RRT), hospitalization for heart failure (HHF), and cardiovascular events within three years after hyperkalemic episodes. The best performing model was further validated using an external cohort. A total of 24,949 adult hyperkalemic patients were selected for model derivation and internal validation. A total of 1452 deaths (16.6%), 887 RRT (10.1%), 1,345 HHF (15.4%), and 621 cardiovascular events (7.1%) were observed. XGB outperformed other models. The area under receiver operator characteristic curves (AUROCs) of XGB vs. LR (95% CIs) for death, RRT, HHF, and cardiovascular events were 0.823 (0.805–0.841) vs. 0.809 (0.791–0.828), 0.957 (0.947–0.967) vs. 0.947 (0.936–0.959), 0.863 (0.846–0.880) vs. 0.838 (0.820–0.856), and 0.809 (0.784–0.834) vs. 0.798 (0.772–0.823), respectively. In the external dataset including 86,279 patients, AUROCs (95% CIs) for XGB were: death, 0.747 (0.742–0.753); RRT, 0.888 (0.882–0.894); HHF, 0.673 (0.666–0.679); and cardiovascular events, 0.585 (0.578–0.591). Kaplan–Meier curves of the high-risk predicted group showed a statistically significant difference from that of the low-risk predicted groups for all outcomes (*p* < 0.005; log-rank test). These findings suggest possible use of machine learning models for real-world risk assessment as a guide for observation and/or treatment decision making that may potentially lead to improved outcomes in hyperkalemic patients while retaining the benefit of life-saving therapies.

## 1. Introduction

Hyperkalemia, characterized by abnormally elevated serum potassium levels, is a common electrolyte abnormality that is often found in patients with heart failure (HF) and chronic kidney disease (CKD) [1,2,3,4,5,6]. The prevalence of hyperkalemia is 2–3% in the general population, whereas notably higher frequencies of hyperkalemia have been reported in patients with diabetes, advanced kidney disease, and those treated with renin-angiotensin-aldosterone inhibitors (RAASi) [7].

Numerous studies have shown that hyperkalemia is associated with increased risks of mortality and adverse clinical outcomes, suggesting the possibility that hyperkalemia can be a marker for the worsening of patients’ general conditions or even the cause of adverse outcomes in certain conditions [8]. Direct and short-term associations between hyperkalemia and mortality risk were reported [9,10,11,12]. Moreover, increased risks of long-term cardiovascular and renal outcomes with a rapid decline of kidney function in hyperkalemic patients were reported [13,14,15], While an association between increased risks of adverse clinical outcomes and hyperkalemia has been well documented, there is limited information on their causality, partially because hyperkalemia is usually multifactorial in its pathogenesis and underlying conditions that can exert influences on patients’ prognoses. For instance, the RAASi treatment discontinuation for reducing the risk of hyperkalemia in HF patients may increase the risk of adverse clinical outcomes [16,17]. Likewise, intense dietary restrictions to mitigate potassium intake for hyperkalemia may lead to the reduced intake of a healthy diet, which in turn may increase the mortality risk of patients with end-stage kidney disease [18,19]. The heterogeneous nature of hyperkalemia makes it difficult to simultaneously assess the risk of different types of adverse clinical events, raising the needs for personalized risk assessment strategies. However, to date, limited evidence is available for conducting risk evaluations of hyperkalemic patients in real-world settings.

Artificial intelligence (AI) in combination with electronic health records has been thought to have the potential to address risks for pathological conditions with heterogenous backgrounds by predicting one-dimensional outcomes based on patients’ multifactorial conditions [20]. In fact, the combination of novel machine learning technology and a high-dimensional real-world database has been shown to be effective for more accurate risk predictions of various diseases compared with conventional statistical risk modeling approaches [20,21]. Thus, it is quite natural to extend the AI approach to personalized risk prediction of hyperkalemic patients with multifactorial conditions.

This study aimed to develop and validate novel AI risk prediction models for hyperkalemic patients with a heterogeneous clinical background using two independent real-world databases. The new machine learning algorithms can assess the risk of mortality and cardiovascular and renal outcomes. The combination of machine learning technology and high-dimensional real-world data has the potential to provide practical predictive accuracy for the personalized detection of hyperkalemic patients at high risk of adverse clinical outcomes and may lead to the improvement of prognosis with more timely and appropriate treatment.

## 2. Materials and Methods

### 2.1. Study Design, Patient Selection, and Data Handling

Data used in this study were extracted from the databases provided by Medical Data Vision Co., Ltd. (MDV; Tokyo, Japan) and Real World Data Co., Ltd. (RWD; Kyoto, Japan). These databases include extensive data on prescriptions, procedures, examinations, laboratory data, and hospital diagnoses based on ICD-10 codes from clinical practice, covering hundreds of medical institutions across most geographic regions and all age groups in Japan. Detailed explanations of these data sources are described in the Appendix A. Models were derived on a retrospective cohort of patients with hyperkalemia extracted from existing hospital records collected in MDV from 1 April 2008 to 30 September 2018.

We selected subjects with hyperkalemia from individuals with at least one serum potassium measurement and aged ≥18 years. Patients with hyperkalemia were defined as those with at least two episodes of elevated serum potassium levels ≥5.1 mmol/L within 12 months. Subjects who were already on dialysis prior to their first hyperkalemic episode, or who had cancer, or who had no history of either HF or CKD stage ≥3a were excluded to ensure substantial patient background homogeneity. The index date was the date of the first hyperkalemic episode, defined as the measurement of serum potassium level ≥5.1 mmol/L. Patients were followed up until the time of death, exit from the dataset or the end of study period, whichever came first.

The data handling flow of the internal dataset is depicted in Appendix A. The dataset of selected patients was divided randomly into two subsets: 80% of patients were included in the model derivation set and 20% were included in the internal validation set. To rigorously assess risk factors of patients, subjects whose hospital records were not available during prior 12 months of the index date were subsequently dropped for model derivation; however, those subjects were retained in the validation set to evaluate the model among broad types of hyperkalemic patients. The derivation set was used to derive the models and cross-validations. For external validation, we selected all patients based on the inclusion and exclusion criteria from the RWD database during the period of 1 January 2009, to 31 December 2019.

### 2.2. Risk Factors and Outcomes

We collected information on medications, medical history, and risk factors based on the information recorded during the 12 months prior to the index date. Risk factors included prescription of RAASi (including angiotensin-converting enzyme inhibitors, angiotensin receptor blockers, and mineralocorticoid receptor antagonists (MRAs)) and other hyperkalemia-inducing drugs, the presence of high-risk conditions (CKD, diabetes mellitus, HF and hypertension), and other comorbidities. We also collected information on laboratory values and typical therapies for hyperkalemia. Variables which were used as predictors for the model are listed in Appendix A. Some data were missed particularly among the laboratory tests (Appendix A). The number of missing data can be retrieved by deducting the observed rate of each variable from the total number of patients (n = 8752) in the derivation set. The handling of missing values for each machine learning algorithm can also be found in the method details of the Appendix A.

The occurrence of clinical outcomes was searched over three years from the first hyperkalemic episode. The studied outcomes were all-cause death, renal replacement therapy introduction (RRT) including dialysis or kidney transplantation, hospitalization for HF (HHF), and cardiovascular events (myocardial infarction, arrhythmia, cardiac arrest, or stroke). Detailed definitions of these outcomes are listed in Appendix A. Furthermore, we exploratorily tested various other types of clinical outcomes listed in Appendix A.

### 2.3. Machine Learning Algorithms

We adopted three types of machine learning algorithms: multiple logistic regression (LR) with L1/L2 regularization [22], extreme gradient boosting (XGB) [23], and deep neural network (NN) [24]. These algorithms were selected based on previous reports showing successful risk predictions using high-dimensional electronical medical record data [25,26]. Models were separately built for the binary classification based on the probability for each clinical outcome; patients were classified as high risk if the probability of the outcome exceeded the pre-determined cut-off points. Detailed explanations of algorithms are described in the Appendix A.

For hyperparameter optimizations of each model, the hyperparameter set that performed best under n-fold cross-validation (5-fold for LR/XGB and 3-fold for NN) was selected (Appendix A). All procedures for model development were implemented using Python 3.7.

### 2.4. Selection of Clinical Variables

Of 81 clinical variables used in the initial models, we selected 64 common variables that can predict the risks of pre-defined clinical outcomes while maintaining the model performance. We took a two-phase approach for the selection. In phase one, we included all the 81 variables, and evaluated prediction accuracy of the models by area under receiver operator characteristics curve (AUROC) within the training dataset. In phase two, we summarized each variable importance by summing the values for all the outcomes and selected candidate variables for deletion according to the following criteria: (1) the rank of summed variable importance lower than 20%, (2) variables that were clinically similar to each other, or (3) variables made of other variables combination. We then evaluated AUROC for the clinical outcomes using experimentally built models by excluding the candidate variables one by one. We finally determined to select the 64 variables set since the prediction accuracy could no longer be maintained when the number of variables was reduced further than the 64 variables (Appendix A).

### 2.5. Validation

The performance of the optimized model was first tested on the internal validation set. For each combination of outcomes and machine learning algorithms, AUROC values, specificity, sensitivity, positive predictive value (PPV), and negative predictive value (NPV) were calculated with the cut-off points of probability of outcomes, i.e., the point maximizing the sum of sensitivity + specificity − 1, herein defined as the best cut-off point. The best cut-off point was set as the cut-off value to the point on the ROC curve farthest from the diagonal line where AUC = 0.5. That is, (sensitivity + specificity − 1) was calculated to obtain the cut-off point that is the maximum value thereof. The point where this (sensitivity + specificity − 1) is the maximum value is defined as the Youden index that provides efficient tradeoff between sensitivity and specificity. An AUROC ≥ 0.80 was considered as an indicator of good prediction performance. To help interpret the instance, Shapley additive explanations (SHAP) values [27] were calculated for each outcome with variables ranked in the top 20 for importance.

The machine learning model which showed the best performance was subjected to external validation, where the model was applied to the external dataset extracted from RWD database. For each outcome, AUROCs, specificity, sensitivity, PPV, and NPV were calculated. The survival curve analysis was carried out based on the probability of clinical outcomes as a threshold. As a result, Kaplan–Meier curves layered in subgroups of high- and low-risk groups were drawn based on the best cut-off points. The survival probabilities between the two groups were verified by the log rank test. Since we could obtain the cause of hospitalization from part of patients (98.4%) in the external validation set, the outcome definitions of HHF and cardiovascular events were modified by counting all hospitalizations with relevant diagnostic codes used to define HF and cardiovascular events (Appendix A). Given limitations of modified definitions for these outcomes, as a *post hoc* analysis, we performed the analysis in a subgroup of patients whose causal information of hospitalization events were available and applied the original definition used in the internal dataset. We also performed another condition of external validation analysis, by restricting the data collection period for input variables within one month after the first hyperkalemic episode for risk predictions of clinical outcomes that occurred after the data collection period, to assess the prediction performance for future adverse events based on the information collected shortly after hyperkalemic episodes. These external validation analyses were performed at an institution independent from the institution performing the model development to ensure reliability of results.

## 3. Results

### 3.1. Patient Selection and Characteristics

Out of 1,208,894 adult patients with at least one serum potassium measurement, we selected 24,949 hyperkalemic patients for model derivation. Among these patients, 4990 patients were held out for the internal validation set; after excluding 11,207 patients whose hospital records were not available during the 12 months prior to the hyperkalemic episode, we selected 8752 patients for the derivation set (Figure 1). For external validation, we selected 86,279 patients from RWD database based on the inclusion and exclusion criteria (Appendix A).

Table 1 shows the patient characteristics for model derivation, internal validation, and external validation sets. Patients included in the derivation and internal validation sets showed similar characteristics with a mean age of 75 years old and 54% males. The mean serum potassium level was 5.4 mmol/L. Approximately 80% and 50–60% of patients had CKD and HF, respectively. Patients included in the external validation set also showed similar age and gender distributions with a mean age of 75 years old and 54% males. The mean serum potassium level was 5.7 mmol/L. 65% and 45% of patients had CKD and HF, respectively.

### 3.2. Model Derivation and Internal Validation

During the study period, 1452 deaths (16.6%), 887 RRT (10.1%), 1345 HHF (15.4%), and 621 cardiovascular events (7.1%) were observed within three years after hyperkalemic episodes in the derivation set. Table 2 presents the prediction performance of XGB, LR, and NN models on the internal validation set with the best cut-off value. A higher prediction performance was obtained in predicting the outcomes with XGB than LR and NN models. The AUROC curves for each model are shown in Figure 2. With XGB, the AUROCs for all outcomes exceeded the threshold for good prediction performance (AUROC ≥ 0.80). The AUROCs of XGB vs. LR for death, RRT, HHF, and cardiovascular events were 0.823 (0.805–0.841) vs. 0.809 (0.791–0.828), 0.957 (0.947–0.967) vs. 0.947 (0.936–0.959), 0.863 (0.846–0.880) vs. 0.838 (0.820–0.856), and 0.809 (0.784–0.834) vs. 0.798 (0.772–0.823), respectively. The results of exploratory outcomes including the prediction performance and AUROC curves are shown in Appendix A.

The best cut-off point was set as the cut-off value to the point on the ROC curve that maximizes the sum of sensitivity + specificity – 1, i.e., the Youden index, that provides efficient tradeoff between sensitivity and specificity.

Figure 3 shows the SHAP summary plots of the top 20 most important variables for XGB. For each type of outcomes, different sets of variables were ranked as variables with high importance. Age, estimated glomerular filtration rate (eGFR), CKD stage, and history of emergency room visit were commonly observed among variables with high importance. Likewise, prescriptions of drugs such as heparin, loop diuretics and sodium bicarbonate, RAASi discontinuation within one year from hyperkalemic episode, and some types of laboratory data including HbA1c, triglyceride, and brain natriuretic peptide commonly appeared among the top 20 most important variables across all outcomes. Compared to LR (Figure 4), XGB considered a broader magnitude of contributions by each clinical variable for risk predictions.

### 3.3. External Validation

Based on the performance evaluation using the internal validation set, XGB was applied to the external validation set. The prediction performances are shown in Table 3. The AUROCs for death, RRT, HHF, and cardiovascular events were 0.747 (0.742–0.753), 0.888 (0.882–0.894), 0.673 (0.666–0.679), and 0.585 (0.578–0.591), respectively (Appendix A). The Kaplan–Meier curves of high- and low-risk groups based on the best cut-off values showed higher incidence of all outcomes in the high-risk group (*p* < 0.005; log-rank test) (Figure 5). When we performed the analysis in a subgroup of patients (n = 84,904) whose causal information of hospitalization events were available and applied the original definitions of outcomes used in the derivation set, the AUROCs for death, RRT, HHF, and cardiovascular events were 0.746 (0.741–0.752), 0.887 (0.881–0.893), 0.784 (0.773–0.796), and 0.636 (0.619–0.652), respectively (Appendix A).

The prediction performances were similar when the data collection period was restricted within one month after hyperkalemic episodes and were used to predict the risk of clinical outcomes that occurred after the data collection period. The AUROCs for death, RRT, HHF, and cardiovascular events were 0.711 (0.704–0.718), 0.867 (0.859–0.874), 0.662 (0.655–0.668), and 0.586 (0.579–0.593), respectively (Appendix A).

## 4. Discussion

We developed and tested the machine learning models for risk predictions of mortality and adverse clinical outcomes over three years after the first hyperkalemic episode. Risk models were built based on multifaceted information obtained from hyperkalemic patients. Among the machine learning models tested, XGB provided the best prediction performance, resulting in AUROCs over 0.8 for all outcomes. The XGB model was further tested on the external validation set and showed that the prediction performances were maintained for death and RRT, but decreased for HHF and cardiovascular events. The high-risk group based on stratification by the machine learning models showed higher incidences for all outcomes.

The prediction models for similar types of outcomes were reported in several studies. A study in patients with HF with preserved ejection fraction showed AUROCs of 0.72 to predict mortality and 0.76 to predict HHF [28]. Another study in dialysis patients showed an AUROC of 0.75 to predict one-year mortality [29]. In our study, the results of internal validation showed prediction performance for death and cardiovascular events were in a similar range, while the prediction performance for RRT and HHF were numerically higher. Although the differences between XGB and LR were not substantial, the XGB models consistently performed better than LR models. Furthermore, the XGB models provided numerically higher sensitivity (recall) and positive predictive values (precision) compared to LR models. These differences could be notable when the model is used for screening patients at high-risk of adverse clinical events. The results of external validation showed the prediction performance of death and RRT were maintained at high levels with some decrease in AUROCs of 0.07–0.08. Considering the models were not optimized for the external dataset, some decreases in prediction performance were within the expected range; however, the decrease of HHF and cardiovascular events were greater by approximately 0.2 AUROCs. This variation could partially be explained by the modified outcome definitions of hospitalization events used in the external validation set. The inclusion of hospitalization events not relevant for HF or cardiovascular events could lead to over estimation of these outcomes in both the high- and low-risk groups, resulting in decreased prediction performance. In fact, the prediction performances were increased by 0.05–0.11 when we applied the original definition of these outcomes to the subset of external validation cohort. These findings suggest that the further performance decline in HHF and cardiovascular events was affected by the modification of outcome definitions.

The important variables shown in the SHAP summary plots suggested that contributions of each clinical variable for risk predictions were identical by outcome types. HF diagnosis, history of emergency visit, high brain natriuretic peptide value, and older age positively related to the risk of HHF, while low eGFR value, advanced CKD stage, history of acute kidney injury, and younger age contributed to the risk of RRT. Likewise, history of cerebrovascular disease, atrial fibrillation or atrial flutter, and myocardial infarction contributed to the risk of cardiovascular events, while older age, history of chronic pulmonary disease, sepsis, and emergency room visit contributed to the risk of death. Interestingly, some variables provided interpretations inconsistent with the clinical knowledge. For instance, our results showed not having RAASi discontinuation after hyperkalemia contributed to the risk of clinical outcomes. Previous studies reported increased risk of adverse clinical outcomes in patients who discontinued RAASi treatment [16,17]. This discrepancy may be explained partially by the fact that the patients not on RAASi treatment at first hyperkalemic episode were also included in the group of “not having RAASi discontinuation” after hyperkalemia. In other words, there were two types of patients including patients with continued RAASi treatment (n = 2566) or those without RAASi treatment (n = 3677) in the population of “not having RAASi discontinuation”. Likewise, the prescription of some medications may not show the risks themselves, but the risk of underlying pathophysiological conditions indicated for such medications. For instance, the prescription of MRA, which is commonly prescribed for the treatment of HF, was selected in the important variables for the risk prediction of HHF. Likewise, treatment by sodium bicarbonate may indicate that patients had metabolic acidosis and were at risk for end-stage kidney disease. Therefore, we must be careful with these interpretations as most predictors were taken from the date of onset of hyperkalemia; and the importance of these variables does not mean the effect of clinical variables as treatments modify the risk of each outcome. The model should merely be used to evaluate the risk of adverse clinical outcomes based on the presented conditions of patients.

Recent studies have shown that hyperkalemic patients are at high risk of long-term cardiovascular and renal outcomes and rapid kidney function decline [13,14]. On the other hand, hyperkalemic patients often have recurrent hyperkalemic episodes [12]. Therefore, treatment of hyperkalemia needs to consider both risks of hyperkalemia and long-term clinical outcomes. However, the treatment for hyperkalemia is complex. Numerous studies have reported that the discontinuation of RAASi treatment to lower serum potassium levels is associated with increased risk of adverse clinical outcomes while reducing the risk of hyperkalemia [16,17]. Likewise, intense dietary restrictions to mitigate potassium intake for hyperkalemia may lead to the reduced intake of a healthy diet, which in turn may increase the mortality risk of patients with end-stage kidney disease [18,19]. In addition, it has been reported that the increased net endogenous acid production (NEAP) by diet was associated with the progression of the renal function decline [30,31]. Since NEAP is an index proportional to protein intake/potassium intake, NEAP can be decreased by high-potassium diet. In fact, several studies reported that the lower CKD risk associated with the high-potassium diet including fruits and vegetables and the increased CKD progression risk associated with low-potassium diets [32,33] or decreased potassium urinary excretion [34,35]. KDOQI clinical practice guideline for nutrition in CKD 2020 Update recommends that reducing NEAP through increased dietary intake of fruits and vegetables in order to reduce the rate of decline of residual kidney function [36]. These data suggest that while potassium restriction can be beneficial for high-risk hyperkalemia patients, intense potassium restriction for low-risk hyperkalemia patients may contribute to higher risk of worsening renal function than risks due to hyperkalemia. Therefore, potassium diet should be carefully guided by considering the risk–benefit balance and each patient’s condition. Nevertheless, it is also true that appropriate dietary guidance based on risk assessment is a very challenging clinical entity because of no indicators, biomarkers, or criteria available for hyperkalemia prognosis at the moment. The developed models provide information on mortality, and cardiovascular and renal outcome risks, within three years after hyperkalemic episodes, which may be used to identify hyperkalemic patients at high risk for adverse clinical outcomes. Therefore, the risk prediction model can play an important role in the personalized risk evaluation of hyperkalemic patients, which enables more proper balancing of the dietary restriction and medical therapy for better clinical outcomes. The treatment of hyperkalemia often leads to the discontinuation or restriction of beneficial but potassium-increasing therapy such as RAASi and high-potassium diet including fruits and vegetables. Based on the personalized risk evaluation of hyperkalemic patients, the AI model aids the mitigation of the adverse event risks of patients while retaining the benefit of these life-saving therapies.

## 5. Strengths and Limitations

One important strength is that the model was tested on an external dataset including more than ten times the number of patients than the derivation set. The prediction performance for the all-cause death, and higher incidence of all clinical outcomes in the high-risk group suggested that the model would work on datasets collected under different settings. However, the prediction performance for HHF and cardiovascular events was decreased, suggesting the need for further attempts to improve the prediction performances. The modifications of outcome definition due to the lack of causal information for hospitalization events could lead to lower prediction performance; therefore, further studies are warranted using clinical outcomes with high specificity applicable across different datasets. Several other approaches may also be considered such as increasing sizes and variety of the training dataset. Advanced technologies such as transfer learning have proven successful to maintain good prediction performance of prediction models across different datasets [37,38]. In this study, we selected the sophisticated machine learning algorithms that have proven effective in previous reports. However, available machine learning algorithms, particularly simpler algorithms, were not comprehensively studied.

The external validation was performed at an independent institution from the institution that performed the model derivation to ensure the reliability of the results. Due to limitations in exchanging the detailed learning conditions, we did not optimize the model in the external dataset; and we did not compare the different machine learning models in the external dataset since the comparison of unoptimized models might not be an ideal condition. However, further studies are needed to externally validate the models in a distinct population or database. Building predictive models using real-world databases had the advantage of large sample sizes representing various clinical settings. However, available information was limited to structured data. Furthermore, MDV collects only deaths that occurred in hospital; therefore, we could not retrieve information on deaths that occurred outside of hospitals. There are some redundancies among the predictor variables used in the final model. For instance, both eGFR value and CKD stage were used as predictor variables. Although we made effort to reduce such redundancies in the variable selection process, they could not be fully removed for maintaining the satisfactory prediction performance of the model. The machine learning modeling algorithm such as the XGB modeling is effective when there are several types of relationships between explanatory variables and objective variables dependent on other variables. Therefore, the application of the machine learning algorithm can aid in the risk prediction based on the numerous types of clinical variables.

Finally, it is important to note that the risk models did not explicitly nor implicitly provide information on the treatment effects of any therapeutic interventions. The treatment of high-risk patients would thoroughly be dependent on existing therapeutic guidelines and assessments by treating physicians based on their clinical knowledge.

## 6. Conclusions

We report the development and validation of risk prediction models using novel machine learning technologies to detect hyperkalemic patients at high risk of mortality, and cardiovascular and renal outcomes, over three years after their first hyperkalemic episode. Although further studies are warranted to improve model applicability in different settings, these findings suggested a possible use of machine learning models for real-world risk assessment as a guide for observation and/or treatment decision making with the potential to lead to the improvement of long-term cardiovascular and renal outcomes, and mortality in patients with hyperkalemia, while retaining the benefit of life-saving therapies.

## Figures and Tables

**Figure 1 nutrients-14-04614-f001:**
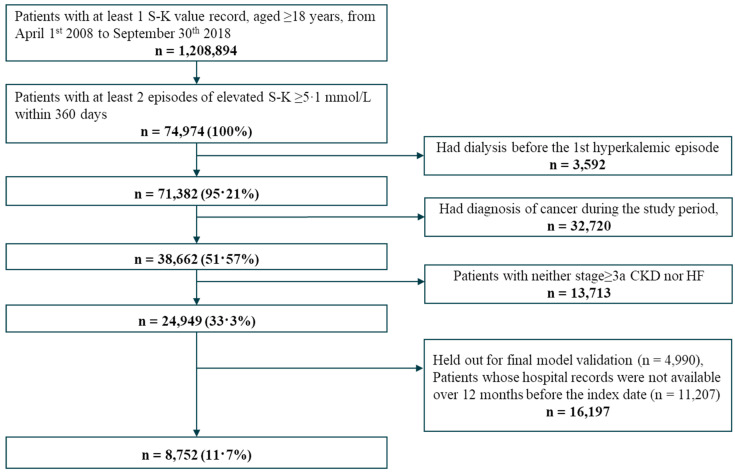
Patient flow diagram for model development.

**Figure 2 nutrients-14-04614-f002:**
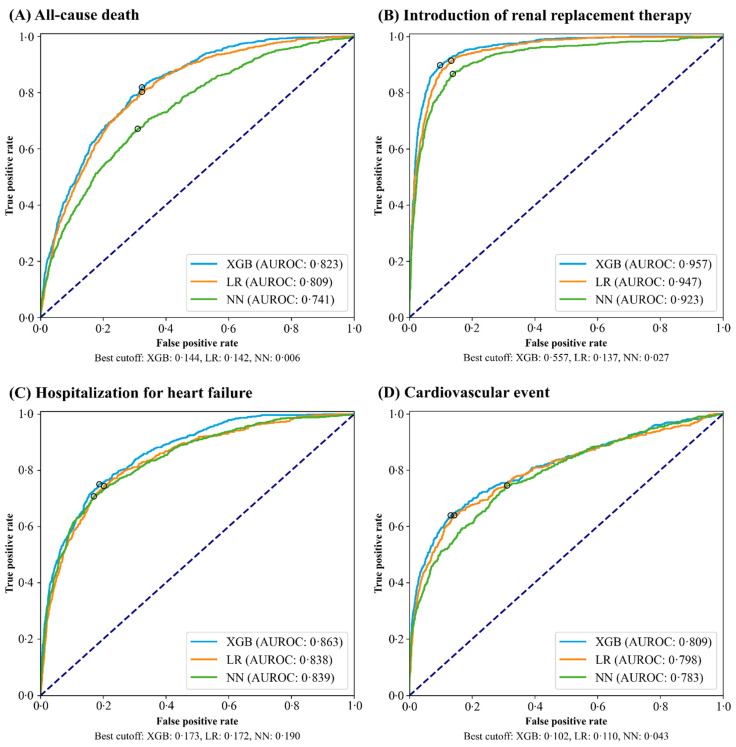
Receiver operator characteristics curves evaluated on the internal validation set. The black rings on each plot show the best-cut off values. XGB, extreme gradient boosting; LR, logistic regression; NN, neural network; AUROC, area under the operator receiver characteristics curve.

**Figure 3 nutrients-14-04614-f003:**
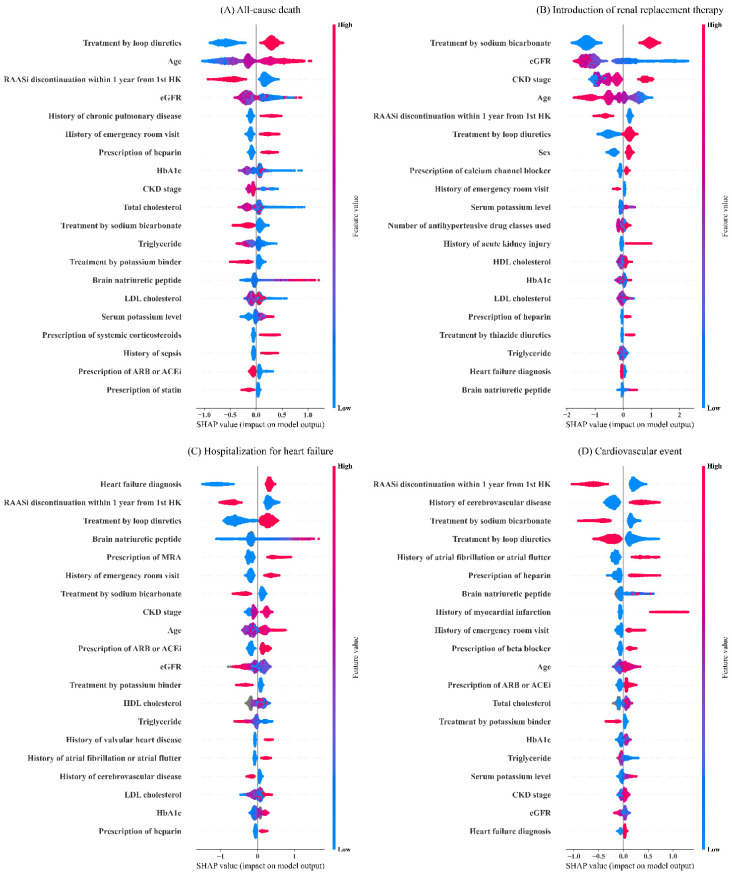
Shapley additive explanations summary plots of the top 20 most important variables for the extreme gradient boosting model.

**Figure 4 nutrients-14-04614-f004:**
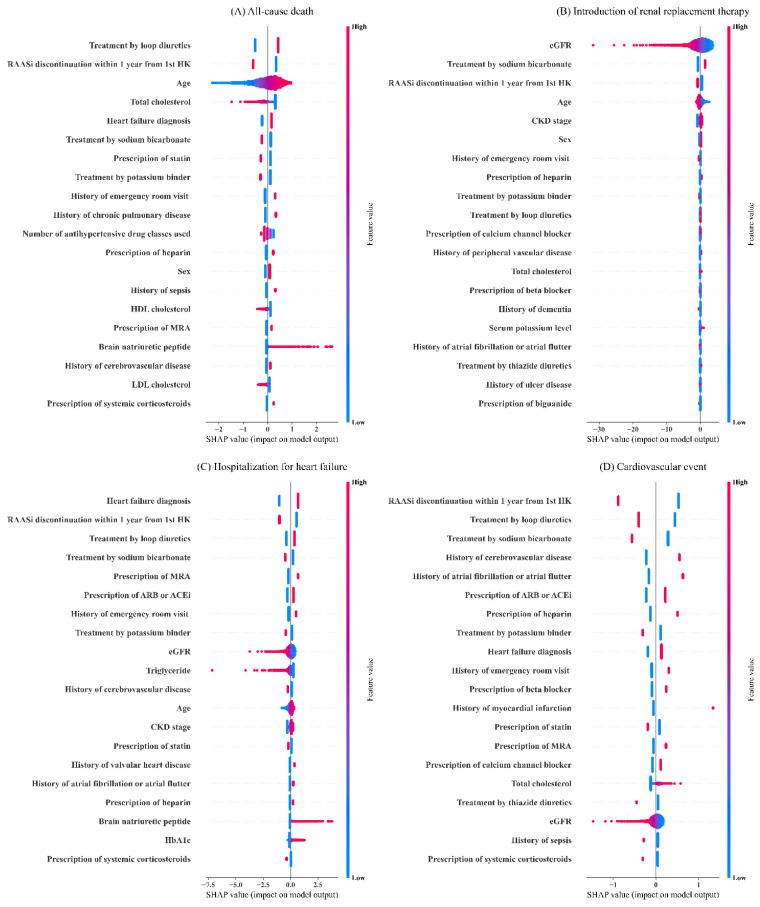
Shapley additive explanations summary plots of the top 20 most important variables for the logistic regression model.

**Figure 5 nutrients-14-04614-f005:**
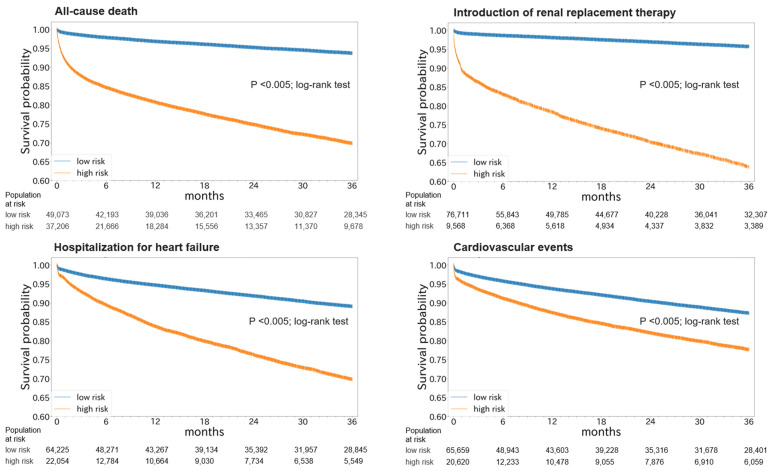
Kaplan–Meier plots of high- and low-risk groups based on risk predictions in the external validation set.

**Table 1 nutrients-14-04614-t001:** Characteristics of patients at the first hyperkalemic episode in the derivation, internal validation, and external validation sets.

	Derivation Set (N = 8752)	Internal Validation Set (N = 4990)	External Validation Set (N = 86,279)
Age (years)			
mean ± SD	75.7 ± 12.4	74.9 ± 12.8	74.9±13.1
Gender, male, n (%)	4717 (53.9)	2697 (54.1)	46,976 (54.4)
Follow up time (days)			
mean ± SD	755.2 ± 641.6	805.3 ± 743.4	664.2 ± 443.3
Serum potassium value (mmol/L)			
mean ± SD	5.4 ± 0.4	5.5 ± 0.5	5.7 ± 2.2
Serum potassium value group, n (%)			
≥5.1 and <5.5 mmol/L	6326 (72.3)	3347 (67.1)	55,984 (64.9)
≥5.5 and <6.0 mmol/L	1727 (19.7)	1094 (21.9)	18,443 (21.4)
≥6.0 and <6.5 mmol/L	434 (5.0)	328 (6.6)	5656 (6.6)
≥6.5 and <7.0 mmol/L	145 (1.7)	123 (2.5)	2492 (2.9)
≥7.0 mmol/L	120 (1.4)	98 (2.0)	3704 (4.3)
CKD, n (%)	6854 (78.3)	4033 (80.8)	56,224 (65.2)
Stage 1	27 (0.4)	11 (0.3)	654 (1.2)
Stage 2	165 (2.4)	80 (2.0)	3771 (6.7)
Stage 3a	1215 (17.7)	628 (15.6)	8607 (15.3)
Stage 3b	1944 (28.4)	1073 (26.6)	12,863 (22.9)
Stage 4	2212 (32.3)	1215 (30.1)	14,570 (25.9)
Stage 5	1,291 (18.8)	1025 (25.4)	15,759 (28.0)
HF, n (%)	5206 (59.5)	2628 (52.7)	38,955 (45.2)
Diabetes, n (%)	4954 (56.6)	2478 (49.7)	31,073 (36.0)
Hypertension, n (%)	7247 (82.8)	3605 (72.2)	31,956 (37.0)
Dyslipidemia, n (%)	3039 (34.7)	1391 (27.9)	17,194 (19.9)
Comorbidity, n (%)			
Myocardial infarction	382 (4.4)	268 (5.4)	5,322 (6.2)
Peripheral vascular disease	1648 (18.8)	798 (16.0)	9,844 (11.4)
Cerebrovascular disease	2,567 (29.3)	1255 (25.2)	13,455 (15.6)
Chronic pulmonary disease	1821 (20.8)	829 (16.6)	8,620 (10.0)
Moderate to severe disease	130 (1.5)	68 (1.4)	868 (1.0)
Atrial flutter or atrial fibrillation	1846 (21.1)	900 (18.0)	9,827 (11.4)
Valvular heart disease	1347 (15.4)	623 (12.5)	8,594 (10.0)
Acute kidney injury	385 (4.4)	309 (6.2)	3,271 (3.8)
Sepsis	1161 (13.3)	537 (10.8)	7,178 (8.3)
Gastrointestinal bleeding	320 (3.7)	178 (3.6)	3,330 (3.9)
Peripheral oedema	343 (3.9)	150 (3.0)	926 (1.1)
eGFR value (mL/min/1.73 m^2^)			
mean ± SD	35.3 ± 22.0	32.9 ± 21.7	37.7 ± 26.6
RAASi treatment, n (%)	5075 (58.0)	2485 (49.8)	30,445 (35.3)
Angiotensin converting enzyme inhibitors	1,041 (11.9)	555 (11.1)	7,629 (8.8)
Angiotensin receptor blockers	3653 (41.7)	1755 (35.2)	21,475 (24.9)
MRA	1,820 (20.8)	881 (17.7)	9,003 (10.4)
Hyperkalemia treatment, n (%)			
Thiazide diuretics	264 (3.0)	122 (2.4)	3,472 (4.0)
Loop diuretics	2186 (25.0)	1251 (25.1)	26,134 (30.3)
Calcium gluconate	181 (2.1)	151 (3.0)	2,587 (3.0)
Sodium bicarbonate	658 (7.5)	402 (8.1)	1,086 (1.3)
Potassium binder (SPS/CPS)	607 (6.9)	404 (8.1)	4,388 (5.1)
Glucose injection and insulin	181 (2.1)	133 (2.7)	972 (1.1)

SD, standard deviation; CKD, chronic kidney disease; HF, heart failure; eGFR, estimated glomerular filtration rate; RAASi, renin-angiotensin-aldosterone system inhibitor; MRA, mineralcorticoid receptor antagonist; SPS, sodium polystyrene sulfonate; CPS, calcium polystyrene sulfonate. The unobserved data (missing data) for laboratory values were imputed in the LR and NN models. The unobserved data for binary variables, such as diagnosis and prescription, responded to negative (the event did not happen), and were set to 0 in all the models.

**Table 2 nutrients-14-04614-t002:** Prediction performance of the machine learning models on the internal validation set.

Outcome	ML Algorithm	AUROC	Sensitivity	Specificity	PPV	NPV
**Cut-off = 0.5**						
All-cause death	XGB	0.823	0.244	0.966	0.594	0.863
	LR	0.809	0.224	0.964	0.556	0.860
	NN	0.741	0.285	0.935	0.470	0.866
Introduction of RRT	XGB	0.957	0.903	0.893	0.594	0.981
	LR	0.947	0.612	0.967	0.761	0.935
	NN	0.923	0.584	0.966	0.750	0.930
Hospitalization for HF	XGB	0.863	0.403	0.967	0.680	0.903
	LR	0.838	0.330	0.967	0.632	0.892
	NN	0.839	0.438	0.948	0.594	0.907
Cardiovascular events	XGB	0.809	0.107	0.998	0.810	0.920
	LR	0.798	0.095	0.996	0.700	0.919
	NN	0.783	0.286	0.982	0.603	0.934
**Best cut-off**						
All-cause death	XGB	0.823	0.819	0.677	0.339	0.949
	LR	0.809	0.802	0.676	0.334	0.944
	NN	0.741	0.670	0.690	0.304	0.912
Introduction of RRT	XGB	0.957	0.899	0.903	0.616	0.981
	LR	0.947	0.914	0.867	0.544	0.983
	NN	0.923	0.866	0.862	0.522	0.974
Hospitalization for HF	XGB	0.863	0.751	0.813	0.411	0.949
	LR	0.838	0.743	0.797	0.389	0.947
	NN	0.839	0.708	0.830	0.420	0.942
Cardiovascular events	XGB	0.809	0.639	0.869	0.320	0.961
	LR	0.798	0.637	0.858	0.302	0.961
	NN	0.783	0.746	0.689	0.189	0.965

ML, machine learning; AUROC, area under the receiver operator characteristic curve; PPV, positive predictive value; NPV, negative predictive value; HF, heart failure; RRT, renal replacement therapy; XGB, extreme gradient boosting; LR, logistic regression; NN, neural network.

**Table 3 nutrients-14-04614-t003:** Prediction performance of the extreme gradient boosting models on the external validation set.

Outcome	AUROC	Sensitivity	Specificity	PPV	NPV
All-cause death	0.747	0.757	0.613	0.209	0.949
Introduction of RRT	0.888	0.555	0.916	0.285	0.971
Hospitalization for HF	0.673	0.445	0.767	0.183	0.922
Cardiovascular events	0.585	0.326	0.771	0.141	0.909

ML, machine learning; AUROC, area under the receiver operator characteristic curve; PPV, positive predictive value; NPV, negative predictive value; HF, heart failure; RRT, renal replacement therapy. Calibration analysis was made based on the best-cut off values. The best cut-off point was set as the cut-off value to the point on the ROC curve that maximizes the sum of sensitivity + specificity − 1, i.e., the Youden index that provides efficient tradeoff between sensitivity and specificity.

## Data Availability

The data included in this manuscript were used under contract with the supplier (Medical Data Vision Co., Ltd. and Real World Data Co., Ltd.) and cannot be freely distributed by the authors.

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
