# Peer review of "Machine Learning Models Predicting Cardiovascular and Renal Outcomes and Mortality in Patients with Hyperkalemia"

_nutrients, 2022, doi:10.3390/nu14214614_

Round 1
Reviewer 1 Report
The study is very interesting and proposes to use an innovative methodology to assess risk in medicine. AI appears very interesting to develop the prediction tool. The study also has a large sample size to develop and validate the models. However, the methodology is unnecessarily complicated and would benefit from simplification.
Major concerns
1. If normokaliemia is defined as a the level of potassium e between 3.5 and 5.0, hyperkaliemia has several cutoff values, 5.5 being the most relevant for clinical outcome. Please justify the cutoff you used.
2. The authors should mention in tables the number of missing data per variable and the way missing values were handled (in the method section, not as a footnote in the table).
3. The cancer patients were excluded. If we evaluate all-cause mortality outcome, why did the authors exclude patients with cancer?
4. The major concern relies on the methodology used that should be simplified. A 10 fold cross validation is more appropriate than the 80/20 percent for internal validation. Using a 10 fold cross validation, the model can be create using 100% of the data. The developed model will then be more appropriated for the external validation.
5. The authors wrote: “To rigorously assess risk factors of patients, subjects whose hospital records were not available during prior 12 months of the index date were subsequently dropped for model derivation. It is not appropriate to use different criteria for the development and internal validation data sets.
6. Why LR/XGB was cross-validated 5 times while NN was cross-validated only 3 times?
7. Presenting prediction performance using 2 different thresholds does not provide any meaningful information. Please provide data using only the best threshold.
8. The number of variables tested and retained to develop the models is unclear to the reviewer and should be clearly defined. A proposed model using 64 variables is not clinically applicable.
9. The reviewer is surprised by the huge difference between Sensitivity and PPV in table 2 (internal validation) and 3 (external validation)
10. Figure 5 should represent only the initial conditions (panels A).
11. It is not reasonable to differentiate between primary and secondary endpoints in this retrospective analysis. Unless specified in clinicaltrials.gov prior to the start of the study.
12. In Table 2, I propose to compare the performance of the 3 models to assess whether there are significant differences between the performance of the models.
13. The discussion needs to be rewritten based on the changes in methodology.
Minor suggestions
1. Authors should justify the use of redundant variables such as creatinine, eGFR, CKD stage.
2. Accuracy is a common assessment for a prediction model. Why didn't the authors specify accuracy in the study?
3. The author may cite this article when reporting the risk of CV outcomes and decreased eGFR and hyperkalemia. Fauvel JP, Gueyffier F, Thijs L, Ducher M. Combined effect of renal function and serum potassium level in sudden cardiac death in aging hypertensive subjects. Hypertens Res. 2018 Jun;41(6):469-474.
Author Response
Oct 20th, 2022
Dr. Lluís Serra-Majem and Dr. Maria Luz Fernandez
Editor-in-Chief
Dear Dr. Serra-Majem,
Dear Dr. Fernandez
Manuscript reference [Nutrients] Manuscript ID: nutrients-1918431
Please find the revised version of our manuscript, entitled “Machine Learning Models Predicting Cardiovascular and Renal Outcomes and Mortality in Patients with Hyperkalemia”, in consideration for publication in Nutrients.
We edited the manuscript carefully in line with the suggestions and comments made by the reviewers, and will enclose two versions of the manuscript, a tracked and clean copy in Word file.
Our responses to the reviewers’ comments are presented in the following pages where the original comments and our responses are described in italic and plain font, respectively. In addition, major modifications in the main text made according to each reviewer`s comment are shown in red in the respective response.
We believe the comments from the reviewers have significantly improved the quality of our manuscript, and we hope that the current revised manuscript with our accompanying responses have properly covered all the points raised by reviewers to be suitable for publication in Nutrients.
Yours sincerely,
Toshitaka Yajima, M.D., Ph.D.
AstraZeneca K.K.
Tower B Grand Front Osaka, 3-1 Ofukacho, Kita-ku, Osaka-shi,
Osaka 530-0011, JAPAN
Email: Toshitaka.Yajima@astrazeneca.com
Tel.: +81-6-4802-3600
Response to the reviewer #1
The study is very interesting and proposes to use an innovative methodology to assess risk in medicine. AI appears very interesting to develop the prediction tool. The study also has a large sample size to develop and validate the models. However, the methodology is unnecessarily complicated and would benefit from simplification.
We would really appreciate for the reviewer’s comments and suggestions that help us substantially improve the manuscript. Below please find our point-by-point responses to each specific comment. In response to the comments from the review, we have supplemented the explanations and rewritten the text in Materials and Methods section to make the methodology more understandable. We hope that our responses adequately answer the main points of reviewer’s comments.
- If normokaliemia is defined as a the level of potassium between 3.5 and 5.0, hyperkaliemia has several cutoff values, 5.5 being the most relevant for clinical outcome. Please justify the cutoff you used.
Response: We appreciate for the reviewer’s comment. Hyperkalemia is typically defined as serum potassium levels that exceed the reference range, and there are no universal criteria for the diagnosis of hyperkalemia. We have adopted the definition used in the previous studies1,2, which we believe is the most broadly taken definition that allows risk prediction for all hyperkalemic patient subgroups with different levels of severity3, including those above 5.5 and 6.0.
- The authors should mention in tables the number of missing data per variable and the way missing values were handled (in the method section, not as a footnote in the table).
Response: We appreciate for the reviewer’s suggestion. The number of missing data per variable was displayed in the table S1 in the supplementary materials. To clarify, we added the explanation in the main text so that the readers will be guided to this information as follows (Page 4 line 125-) ;
“There are missingness of the data particularly among the laboratory tests (Table S1-(a)). The number of missing data can be retrieved by deducting the observed rate of each variable from the total number of patients (n = 8,752) in the derivation set. The handling of missing values for each machine learning algorithms can also be found in the method details of the supplementary materials.”
- The cancer patients were excluded. If we evaluate all-cause mortality outcome, why did the authors exclude patients with cancer?
Response: Thank you very much for the reviewer’s comment. In this study, we aimed to study the association between hyperkalemia and the risk of adverse clinical events. When assessing outcomes for all-cause mortality, patients with cancer should be included. However, in this study, we aimed to study the association between hyperkalemia and the risk of adverse clinical events. Patients with cancer may die mainly of cancer and the mortality risk may be dependent on the pathophysiological conditions/ progression status of the cancer. Therefore, we excluded the cancer patients to exclude the effect of the cancer-associated mortality risk in the analysis.
- The major concern relies on the methodology used that should be simplified. A 10 fold cross validation is more appropriate than the 80/20 percent for internal validation. Using a 10 fold cross validation, the model can be create using 100% of the data. The developed model will then be more appropriated for the external validation.
Response: We appreciate for the reviewer’s careful review and suggestion. While the alternative approach such as a 10-fold cross validation can be used by fully utilizing the internal dataset, the 80/20 data split for the model derivation and testing (internal validation) is also a commonly employed approach for the development of machine-learning-based risk prediction models4. The purpose of dividing the data set into the derivation and testing is to improve generalization performance (performance on unknown data). If all data is used for training, the risk of over-learning (a state in which only training data is suitable, and answers cannot be output accurately to untrained data) cannot be eliminated. Even though the available sample size for our model derivation was reduced by the data split, we eventually showed satisfactory prediction performances exceeding pre-defined threshold of AUROC >0.80 for all main outcomes. Therefore, we believe that the approach we have taken was also practically applicable for the model building.
- The authors wrote: “To rigorously assess risk factors of patients, subjects whose hospital records were not available during prior 12 months of the index date were subsequently dropped for model derivation. It is not appropriate to use different criteria for the development and internal validation data sets.
Response: Thank you very much for the reviewer’s comment. For the model development, we considered it was important to rigorously collect the clinical information of patients for the risk prediction. Therefore, we dropped the subjects whose hospital records were not available during prior 12 months from the model derivation set. We think this approach is appropriate because the model derivation and the internal validation process were clearly distinguished nevertheless the sample populations of both datasets were essentially the same other than the availability of the criteria that necessitates the availability of pre-index hospital records. Moreover, the condition of the internal validation is assumed to be disadvantageous for the developed model where the risk factor information was missing for some patients. Even in such conditions, the models demonstrated good prediction performances. These results adequately suggest the validity of the developed machine learning models.
- Why LR/XGB was cross-validated 5 times while NN was cross-validated only 3 times?
Response: It was necessary to secure the multiple attempts for the cross validation to assess the model performance. In this regard, we were able to secure at least 3-time cross-validation for all the algorithms. The reason for a smaller number of times for the cross validation of NN was that the NN modeling was computationally intensive and required much longer time for calculation than other algorithms.
- Presenting prediction performance using 2 different thresholds does not provide any meaningful information. Please provide data using only the best threshold.
Response: We appreciate for the reviewer’s comment to improve the readability of the results. The table 2 and corresponding explanation in the main text have been revised to include only the results of best cut-off value
Changes made in the main text:
Page 5 line 157~
(Before) The performance of the optimized model was first tested on the internal validation set. For each combination of outcomes and machine learning algorithms, area under receiver operator characteristics curve (AUROC) values, specificity, sensitivity, positive predictive value (PPV), and negative predictive value (NPV) were calculated with two cut-off points of probability of outcomes i.e., a fixed value 0.5 and the point maximizing the sum of sensitivity + specificity – 1, herein defined as the best cut-off point.
(After) The performance of the optimized model was first tested on the internal validation set. For each combination of outcomes and machine learning algorithms, AUROC values, specificity, sensitivity, positive predictive value (PPV), and negative predictive value (NPV) were calculated with the cut-off points of probability of outcomes i.e., the point maximizing the sum of sensitivity + specificity – 1, herein defined as the best cut-off point.
Page 8 line 245~
(Before) Although the differences between XGB and LR were not substantial, the XGB models consistently performed better than LR models. Furthermore, the XGB models provided consistently higher sensitivity (recall) and positive predictive values (precision) with the fixed cut-off value (probability) 0.5.
(After) Although the differences between XGB and LR were not substantial, the XGB models consistently performed better than LR models. Furthermore, the XGB models provided numerically higher sensitivity (recall) and positive predictive values (precision) compared to LR models.
Table2: We moved the model performance based on cut-off value = 0.5 for more simplified and clearer results presentation
- The number of variables tested and retained to develop the models is unclear to the reviewer and should be clearly defined. A proposed model using 64 variables is not clinically applicable.
Response: Thank you very much for the reviewer’s comment. The model was initially derived using 81 clinical variables. Through the variable selection process as described in the method section, we finally selected the 64 common clinical variables for the risk prediction of four main outcomes. We have intensively modified the texts in “Selection of clinical variables” in the materials and methods for more clarity as described below. As to the real-world clinical applicability you pointed out, we think it is possible to extract the variables from medical records automatically through the appropriate system development since all the clinical variables used were defined based on the structured clinical information. In fact, we have already started developing such system though it is still in a preliminary phase.
Paragraph changed in the main text: Page 5 line 145~
Selection of clinical variables
Of 81 clinical variables used in the initial models, we selected 64 common variables that can predict risks of pre-defined clinical outcomes while maintaining the model performance. We took a two-phase approach for the selection. In phase one, we included all the 81 variables, and evaluated prediction accuracy of the models by area under receiver operator characteristics curve (AUROC) within the training dataset. In phase two, we summarized each variable importance by summing the values for all the outcomes and selected candidate variables for deletion according to the following criteria; 1) the rank of summed variable importance lower than 20%, 2) variables that were clinically similar to each other, or 3) variables made of other variables combination. We then evaluated AUROC for the clinical outcomes using experimentally built models by excluding the candidate variables one by one. We finally determined to select the 64 variables set since the prediction accuracy could no longer be maintained when the number of variables was reduced further than the 64 variables. (Table S2).
- The reviewer is surprised by the huge difference between Sensitivity and PPV in table 2 (internal validation) and 3 (external validation)
Response: Since the sensitivity and PPV results can be changed by altering the cut-off value to distinguish the patients at high- or low-risk, it is more appropriate to compare the AUROC values to measure the model performance in the two different datasets. We discussed in the main text about the expected reasons for the differences in the model performance as follows (Page 8 line 251~);
“Considering the models were not optimized for the external dataset, some decreases in prediction performance were within the expected range; however, the decrease of HHF and cardiovascular events were greater by approximately 0.2 AUROCs. This variation could partially be explained by the modified outcome definitions of hospitalization events used in the external validation set. The inclusion of hospitalization events not relevant for HF or cardiovascular events could lead to over estimation of these outcomes in both the high- and low-risk groups, resulting in decreased prediction performance. In fact, the prediction performances were increased by 0.05–0.11 when we applied the original definition of these outcomes to the subset of external validation cohort. These findings suggest that the further performance decline in HHF and cardiovascular events was affected by the modification of outcome definitions.”
- Figure 5 should represent only the initial conditions (panels A).
Response: We appreciates for the reviewer’s suggestion. In accordance with the comment, we moved the results of the restricted condition (panel B) from the Fig 5 and Table 5 to the supplementary Fig S6 and Table S6 for more simplified and clearer results presentation for readers.
- It is not reasonable to differentiate between primary and secondary endpoints in this retrospective analysis. Unless specified in clinicaltrials.gov prior to the start of the study.
Response: We would appreciate for the reviewer for pointing this out. We understand the reviewer’s comment and revised the main text by excluding the phrase “primary” and “secondary” to explain the main outcomes.
Page 8 line 129~
(Before) The primary outcome was all-cause death, and the secondary outcomes included renal replacement therapy introduction (RRT) including dialysis or kidney transplantation, hospitalization for HF (HHF), and cardiovascular events (myocardial infarction, arrhythmia, cardiac arrest, or stroke)
(After) The studied outcomes were all-cause death, renal replacement therapy introduction (RRT) including dialysis or kidney transplantation, hospitalization for HF (HHF), and cardiovascular events (myocardial infarction, arrhythmia, cardiac arrest, or stroke).
- In Table 2, I propose to compare the performance of the 3 models to assess whether there are significant differences between the performance of the models.
Response: We thank for the reviewer’s comment. While it is of interest to assess the statistical significance of the differences in the model performances, the purpose of the model derivation and the internal validation was to select the best performing modeling algorithm based on the multifaceted aspects. In this study, we focused on the consistency in the trend of the model performance across the main outcomes rather than the difference among the modeling algorithms to select the best performing model. In addition, we thought that showing the statistical significance may mislead the readers in terms of the purpose of the model derivation process. Therefore, we decided not to include this information. However, we would like to consider including the statistical significance in the performance of the 3 models in the other occasion when appropriate.
- The discussion needs to be rewritten based on the changes in methodology.
Response: We thank the reviewer’s careful review and suggestions. In accordance with the guidance in each comment, we revised the main text of the manuscript.
Specifically, we made following changes:
- Correcting the explanation about the results based on the cut-off value 0.5, in accordance with the reviewer’s suggestion for showing only the results based on the best cut-off value.
- Revised the main text explaining the results of restricted condition for the external validation in accordance with the move of main figures/tables of these results to the supplementary figures/tables.
Changes made in the main text:
Page 5 line 157~
(Before) The performance of the optimized model was first tested on the internal validation set. For each combination of outcomes and machine learning algorithms, area under receiver operator characteristics curve (AUROC) values, specificity, sensitivity, positive predictive value (PPV), and negative predictive value (NPV) were calculated with two cut-off points of probability of outcomes i.e., a fixed value 0.5 and the point maximizing the sum of sensitivity + specificity – 1, herein defined as the best cut-off point.
(After) The performance of the optimized model was first tested on the internal validation set. For each combination of outcomes and machine learning algorithms, AUROC values, specificity, sensitivity, positive predictive value (PPV), and negative predictive value (NPV) were calculated with the cut-off points of probability of outcomes i.e., the point maximizing the sum of sensitivity + specificity – 1, herein defined as the best cut-off point.
Page 8 line 245~
(Before) Although the differences between XGB and LR were not substantial, the XGB models consistently performed better than LR models. Furthermore, the XGB models provided consistently higher sensitivity (recall) and positive predictive values (precision) with the fixed cut-off value (probability) 0.5.
(After) Although the differences between XGB and LR were not substantial, the XGB models consistently performed better than LR models. Furthermore, the XGB models provided numerically higher sensitivity (recall) and positive predictive values (precision) compared to LR models.
Table2: We moved the model performance based on cut-off value = 0.5 for more simplified and clearer results presentation
- Authors should justify the use of redundant variables such as creatinine, eGFR, CKD stage.
Response: We appreciate for the reviewer’s comment. Considering that the CKD stage is determined based on the eGFR values, we understand that there is a certain redundancy among the predictor variables in the model. It was difficult to adjust all redundancy among the predictor variables while maintaining the model performance. Therefore, some of the redundancy was not fully removed in the final model. As we explained in the material and methods section, we made efforts to reduce the redundancy among the risk factors in the variable selection process. In this process, we removed the predictor variables with a small importance for the risk predictions and those assumed to be redundant with other variables. In this process, the candidate variables for deletion were determined based on the variable importance values as well as those assumed to be clinically similar to or combination of other variables. We also assessed the prediction performance of the model after deletion of candidate variables. Since the model performance of the risk prediction was also an important measure to guide the variable selection, the possibility of including some redundant variables could not be fully denied such as those variables that the reviewer commented. However, the machine learning modeling algorithms such as the XGB modeling is effective when there are several types of relationships between explanatory variables and objective variables dependent on other variables. Therefore, the application of the machine learning algorithm can aid in justifying the risk prediction based on the numerous types of clinical variables where the redundancy among predictor variables cannot be fully excluded.
According to the reviewer`s comment, we have described a justification of including redundant variables in the main text as follows (Page 10 line 332~); “There are some redundancies among the predictor variables used in the final model. For instance, both eGFR value and CKD stage were used as predictor variables. Although we made effort to reduce such redundancies in the variable selection process, they could not be fully removed for maintaining the satisfactory prediction performance of the model. The machine learning modeling algorithm such as the XGB modeling is effective when there are several types of relationships between explanatory variables and objective variables dependent on other variables. Therefore, the application of the machine learning algorithm can aid in the risk prediction based on the numerous types of clinical variables.”
- Accuracy is a common assessment for a prediction model. Why didn't the authors specify accuracy in the study?
Response: The reason for employing the sensitivity, specificity, PPV, and NPV for the model performance was that those measures are commonly used in the clinical research. Nevertheless, we assessed AUROC, which is commonly used as the performance metrics, as the primary measure of the performance of risk predictions for the model selection and the validation analyses.
- The author may cite this article when reporting the risk of CV outcomes and decreased eGFR and hyperkalemia. Fauvel JP, Gueyffier F, Thijs L, Ducher M. Combined effect of renal function and serum potassium level in sudden cardiac death in aging hypertensive subjects. Hypertens Res. 2018 Jun;41(6):469-474.
Response: We appreciate for the reviewer’s suggestion. Based on the comment, this article has been referenced in the main text (Page 3 line 70) as described below with the citation.
“Moreover, increased risks of long-term cardiovascular and renal outcomes with a rapid decline of kidney function in hyperkalemic patients were reported. [Ref 15]”
Response to the reviewer #2
The paper entitled: "Machine Learning Models Predicting Cardiovascular and Renal Outcomes and Mortality in Patients with Hyperkalemia" is a study to evaluate machine learning to develop predictive models in hyperkalemic patients. Risk prediction models were assessed using extreme gradient boosting (XGB), multiple logistic regression (LR), and deep neural network. The authors concluded that machine learning allows for detecting hyperkalemic patients at high risk of mortality and cardiovascular and renal outcomes.
Response: We would really appreciate for the reviewer’s insightful comments and suggestions. Based on the comments and suggestion provided, we made a revision in the manuscript to improve the clarity for readers. Below please find our point-by-point responses to each comment. We hope that our responses would adequately answer the main points of the reviewer’s comment.
- The authors defined hyperkalemia as a serum value above 5.1 mmmol/l. This definition of hyperkalemia is not universally accepted, and I suggest assessing the predictive model using 5.5 mmol/L as the threshold to determine hyperkalemia.
Response: We appreciate for the reviewer’s comment. Hyperkalemia is typically defined as serum potassium levels that exceed the reference range, and there are no universal criteria for the diagnosis of hyperkalemia. We have adopted the definition used in the previous studies2,3, which we believe is the most broadly taken definition that allows risk prediction for all hyperkalemic patient subgroups with different levels of severity1, including those above 5.5 and 6.0.
- The authors should clarify why they selected patients with two episodes of hyperkaliemia but chose the date of the first episode as "index date". If they used two measurements to define hyperkalemia, they should use the date of the second measurement because the conditions (e.g.: eGFR decline) between the two assessments could be changed.
Otherwise, they could consider evaluating patients with a single episode of hyperkalemia to reduce selection bias.
Response: As we mentioned in the response to the previous comment, the definition using two serum potassium values above 5.1 mmol was based on the definition use in the previous study.2,3 In the present study, we aimed to elaborate the risk prediction model for long-term cardiovascular and renal outcomes in hyperkalemic patients. Therefore, the study intended to include the patients likely to have chronic hyperkalemia due to decreased potassium elimination because of advanced kidney disease and/or the use of RAASis to treat underlying kidney or heart conditions. Furthermore, the definition based on two measurements of potassium would avoid inclusion of patients with elevated potassium due to testing errors, e.g., pseudo-hyperkalemia. While we used the two potassium measurements to rigorously select the hyperkalemic patients, it was also important to secure the sufficient observation period since models were built to predict the long-term risks (i.e., occurred in 3-year) of cardiovascular and renal outcomes. Therefore, we set the index date as the first hyperkalemic episode to secure the longer observational period in each patient. The needs of second measurement may lead certain types of bias; however, this type of bias could be mitigated as far as the cohort data is used to build the risk prediction model based on the risk factors appropriately collected in the same cohort and not to be used to compare the outcomes with the other population.
- In both cases, there is a concern, but I believe that if the authors select only one serum potassium measurement >5.5 mmol/L, they could obtain "true" hyperkalemic patients. Then, they could collect outcomes from the time of first hyperkalemia, avoiding misinterpretation due to time-dependent changes of those factors affecting serum levels (GFR variation, RAASi discontinuation, insulin therapy).
Response: There are pros and cons for using one serum potassium measurement >5.5 mmol/L to select the hyperkalemic patients. While indexing with the single point of measurement can mitigate the influence of time-dependent changes of certain types of risk factors, the higher cut-off value leads to the reduction of sample size eligible for the study. As shown in the table 1, approximately 25–30% of patients had serum potassium value ≥5.5 mmol/L at index date. The substantial reduction in sample size may limit the applicability of machine learning algorithm to build the risk prediction model due to the reduced number of observed clinical events. Furthermore, the definition based on a single measurement of potassium may not exclude the possibility of including patients with pseudo hyperkalemia, e.g., testing errors or hemolysis in the study. Although, we would not be able to determine whether the suggested criteria (serum potassium measurement >5.5 mmol/L) or our approach were the better than the others, we would like to take the reviewer’s comment in the future study.
- Furthermore, it could be helpful to include a dichotomic variable (RAASi use yes vs no ) to avoid misinterpretation of the findings regarding the discontinuation of these drugs.
Response: We appreciate for the reviewer’s comment. Although the variable RAASi use was not selected in the final model, these dichotomic variables were considered when developing the model. In fact, we included the variable “RAASi use” as a predictor variable in the initial model before the variable selection step. In this model, we also used the “use of ACEi or ARB” and the “use of MRA.” Next, in the variable selection step, we reduced the number of predictor variables used in the initial model to reduce the redundancy of the variables and variables with small importance for risk prediction. After careful assessment of the changes in the model performance by the removal of the candidate variables, we reached to the 64 variables in the final models from the 81 clinical variables used in the initial model. The use of RAASi was removed in this variable selection step based on the process described in the method section.
- I noticed that deep neural network was inferior to logistic regression. How do the authors explain this finding?
Response: We thank the reviewer for pointing this out. The good prediction performance of XGB and LR may indicate that non-linearity in data is not very significant. In previous research, an ensemble learning method and some regression-based methods represented higher performance than neural network or support vector machine to classify lung cancer severity using healthcare claims dataset.4 The other systematic review reported that NN does not always show superior prediction performance compared to other machine learning algorithms.5 For dataset from healthcare records, there might be a relatively large linear relationship between available parameters and clinical outcomes. Therefore, NN may suffer from overfitting. While NN is efficient for dealing with unstructured data such as image, video, text and audio, the question of whether NN can also perform the best on structured data still remain.6
- Is there some information about HK- related ECG alterations (e.g., peaked T waves, P wave widening/flattening, PR prolongation)? According to the authors, could implementing this information could improve the predictivity of the machine learning models?
Response: Thank you very much for pointing this out. However, these variables were not available in the dataset used in this study. HK-related ECG alterations such as peaked T waves, P wave widening/flattening, and PR prolongation may have a potential to improve the predictability of certain clinical outcomes, e.g., all-cause mortality and cardiac events. Therefore, we would like to consider these variables for the improvement of model performance in the future study.
- Overall, I struggled with reading this article. Therefore, I suggest an extensive review to clarify the clinical message and the utility of these novel methods in the real clinical world. For instance, extend the legend of Figure 4 and figure 5 to improve their interpretation.
Response: The study aimed to elaborate the risk prediction model for hyperkalemic patients with a heterogeneous clinical background. As explained in the Introduction, some interventions to lower the serum potassium level led to a contradictory short-term effect by reducing the risk of sustained or recurrent hyperkalemia while increasing the risk of adverse cardiovascular outcomes. Therefore, the use of individualized risk evaluation on cardiovascular and renal outcomes and mortality in this population would aid in determining the observation/management of hyperkalemia while considering both the risk of long-term clinical outcomes and the risk of hyperkalemia.
As the reviewer commented, we learned that the important variables shown in the SHAP summary plots aid the interpretation of individualized risk evaluation in this heterogenous cohort of hyperkalemic patients. Therefore, we kept the space to discuss the insight obtained from the SHAP summary plot analysis in the discussion part (Page 8 line 260~: The important variables shown in the SHAP summary plots...).
Furthermore, in the later part, we discussed about the potential clinical benefit from the individualized risk evaluation for hyperkalemic patients (Page 9 line 282~: Recent studies have shown that...; line 302~: The develop models provide information on mortality, and cardiovascular and renal outcomes risks... which may be used to...).
We also have added clinical message and the utility in the introduction part
Added: (Page 3 line 92) The combination of machine learning technology and high-dimensional real-world data has the potential to provide practical predictive accuracy for the personalized detection of hyperkalemic patients at high risk of adverse clinical outcomes and may lead to the improvement of prognosis with more timely and appropriate treatment.
With these information, we hope to provide the meaningful insights based on the study for the machine-learning-based risk evaluation for cardiovascular and renal outcomes of hyperkalemic patients with multifactorial conditions.
Reference
- Kashihara N, Kohsaka S, Kanda E, et al. Hyperkalemia in real-world patients under continuous medical care in Japan. Kidney Int. Rep. 2019; 4(9):1248-1260.
- Betts KA, Woolley JM, Mu F, et al. The prevalence of hyperkalemia in the United States. Curr Med Res Opin. 2018; 34: 971-978.
- Kovesdy CP. Management of hyperkalemia in chronic kidney disease. Nat Rev Neprol. 2014; 10: 653-662.
- Giulia Preti M, Baglio F, Marcella L, et al. Assessing Corpus Callosum Changes in Alzheimer's Disease: Comparison between Tract-Based Spatial Statistics and Atlas-Based Tractography. PLOS ONE. 2015;7(4): e35856.
- Bergquist SL, Brooks GA, Keating NL, Landrum MB, Rose S. Classifying Lung Cancer Severity with Ensemble Machine Learning in Health Care Claims Data. Proc Mach Learn Res. 2017; 68: 25-38.
- Christodoulou E, Ma J, Collins GS, et al. A systematic review shows no performance benefit of machine learning over logistic regression for clinical prediction models. J Clinical Epidemiol. 2019; 110: 12-22.
- Kwak, G. H., & Hui, P. DeepHealth: Review and challenges of artificial intelligence in health informatics. arXiv preprint. 2019;1909.00384.

Reviewer 2 Report
The paper entitled: "Machine Learning Models Predicting Cardiovascular and Renal Outcomes and Mortality in Patients with Hyperkalemia" is a study to evaluate machine learning to develop predictive models in hyperkalemic patients. Risk prediction models were assessed using extreme gradient boosting (XGB), multiple logistic regression (LR), and deep neural network. The authors concluded that machine learning allows for detecting hyperkalemic patients at high risk of mortality and cardiovascular and renal outcomes.
Following are some considerations:
The authors defined hyperkalemia as a serum value above 5.1 mmmol/l. This definition of hyperkalemia is not universally accepted, and I suggest assessing the predictive model using 5.5 mmol/L as the threshold to determine hyperkalemia.
The authors should clarify why they selected patients with two episodes of hyperkaliemia but chose the date of the first episode as "index date".
If they used two measurements to define hyperkalemia, they should use the date of the second measurement because the conditions (e.g.: eGFR decline)between the two assessments could be changed.
Otherwise, they could consider evaluating patients with a single episode of hyperkalemia to reduce selection bias.
In both cases, there is a concern, but I believe that if the authors select only one serum potassium measurement >5.5 mmol/L, they could obtain "true" hyperkalemic patients. Then, they could collect outcomes from the time of first hyperkalemia, avoiding misinterpretation due to time-dependent changes of those factors affecting serum levels (GFR variation, RAASi discontinuation, insulin therapy).
Furthermore, it could be helpful to include a dichotomic variable (RAASi use yes vs no ) to avoid misinterpretation of the findings regarding the discontinuation of these drugs.
I noticed that deep neural network was inferior to logistic regression. How do the authors explain this finding?
Is there some information about HK- related ECG alterations (e.g., peaked T waves, P wave widening/flattening, PR prolongation)? According to the authors, could implementing this information could improve the predictivity of the machine learning models?
Overall, I struggled with reading this article. Therefore, I suggest an extensive review to clarify the clinical message and the utility of these novel methods in the real clinical world. For instance, extend the legend of Figure 4 and figure 5 to improve their interpretation.
Author Response

(The authors gave the same response as above.)
